# Wheat Space Odyssey: “From Seed to Seed”. Kernel Morphology

**DOI:** 10.3390/life9040081

**Published:** 2019-10-25

**Authors:** Ekaterina N. Baranova, Margarita A. Levinskikh, Alexander A. Gulevich

**Affiliations:** 1Department of Plant Cell and Genetic Engineering, All-Russian Research Institute of Agricultural Biotechnology, Moscow 127550, Russia; a_gulevich@mail.ru; 2Department Institute of Biomedical Problems, Moscow 123007, Russia; r.levinskikh@gmail.com

**Keywords:** plants in space, super-dwarf wheat, morphology of wheat grains, grain cover, life support systems, international space station

## Abstract

The long-term autonomous existence of man in extraterrestrial conditions is associated with the need to cultivate plants—the only affordable and effective means for both providing oxygen and CO_2_ utilization, and providing one of the most habitual and energetically valuable products: plant food. In this study, we analyzed the results of the space odyssey of wheat and compared the morphological features of parental grains harvested from soil grown wheat plants, the grains obtained from plants grown in a specialized device for plant cultivation—the “Lada” space greenhouses during space flight in the ISS, and the grains obtained from plants in the same device on Earth. The seeds obtained under various conditions were studied using scanning electron microscopy. We studied the mutual location of the surface layers of the kernel cover tissues, the structural features of the tube and cross cells of the fruit coat (pericarp), and the birsh hairs of the kernels. It was found that the grains obtained under wheat plants cultivation on board of the ISS in near space had some specific differences from the parental, original grains, and the grains obtained from plants grown in the “Lada” greenhouse in ground conditions. These changes were manifested in a shortening of the birsh hairs, and a change in the size and relative arrangement of the cells of the kernel coat. We suggest that such changes are a manifestation of the sensitivity of the cytoskeleton reorganization systems and water exchange to the influence of particular physical conditions of space flight (microgravity, increased doses of radiation, etc.). Thus, the revealed changes did not hinder the wheat grains production “from seed to seed”, which allows the cultivation of this crop in stable life support systems in near earth orbit.

## 1. Introduction

The cultivation of cereals is crucial for the development of human civilization in the transition from gathering to settled agriculture. The first examples of the cultivation of diploid and tetraploid wheat were noted in south-eastern Turkey [1]. Further, the cultivation spread to the territory of the Middle East already in the form of hexaploid bread wheat, which is grown everywhere [2]. Finally, the question arose of promoting this crop along with the expansion of the human habitat. So far, man has made only timid steps beyond the limits of the earth’s atmosphere, and is only beginning to explore outer space aboard various spacecrafts and long-term orbital stations. The growing of dicotyledonous plants in near Earth orbit in the Soviet (Russian) “Mir” space station and the International Space Station (ISS) was quite stable [3]. Obtaining seeds from plants such as *Brassica rapa* L., *Arabidopsis thaliana* L., *Pisum sativum* L. during orbital flight has become a routine [4,5,6]. It is not surprising that wheat was chosen from monocotyledonous plants for the experiments on space station: it is one of the most important crops for human civilization. Owing to the limitation of the ISS volume for experimental plant growth system in space flight, a specially obtained dwarf form of wheat was chosen, the progenitors of which provided one of the most impressive agricultural successes during the green revolution due to the creation of super-effective short-stemmed cultivars. However, for a number of reasons related to the relatively large size of plants, the peculiarities of the gas composition in space inhabited modules (especially because of the presence of excess ethylene), for a long time it was not possible to obtain seeds of this valuable plant [7]. The great achievement was that it was possible to get the seeds after cultivating the plants in space, returning them to Earth and growing them up to the stage of maturation of the grains [8]. These experiments revealed the difficulty of wheat cultivation and the high demands of this crop on the content of ethylene, which did not interfere with the flowering of plants, but caused disturbances in the generative sphere, leading to impaired fertility [9]. More recently, scientists and cosmonauts together were able to overcome these obstacles and get the first wheat seeds in space flight conditions. For the first time, successful cultivation of wheat from seed to seed was carried out by cosmonaut Maxim Suraev as part of the Rasteniya-2 experiment during the Expedition 21/22 on ISS. Repeated experiments made it possible to specify a number of cultivation parameters, and to obtain a sufficient number of samples for various experiments [10]. For this work, it was necessary to create and improve: (1) special dwarf forms by producing hybrids with other varieties of wheat (*Triticum sphaerococcum* Perc.), which made it possible to create a super-dwarf form that was used in a limited space of plant vegetation module [11]; (2) the engineering of a new plant growth system—the space greenhouse “Lada” [10]; (3) inclusion of experiments in the scientific program of missions on the ISS [12]. Currently, the experiment has been successfully repeated: wheat has been grown in the new Spaceflight Plant Growth Systems [13].

The primary visual analysis and growing of the offspring of seeds obtained in near space, as expected, did not reveal significant deviations in the shape and quantity of grains, their germination and viability [6]. However, a lot of data established at the cellular level for eukaryotic cells (changes in the cytoskeleton, in a number of metabolic reactions) [14,15,16] suggests that subtle changes can be detected by more detailed studies. Various compact greenhouses engineered for cultivating of plants aboard spacecrafts and stations are characterized by altered growing conditions, which are different from conditions of growing in soil. These conditions can significantly affect the formation of both the root system and the whole plant. The cytoskeleton is a sensitive system and responds to the adverse effects of abiotic factors, causes changes in cell expansion, and also affects the direction of cell division [17,18]. The observed changes in the cell wall and its surface morphology (and in the surface of coat tissues, accordingly) are associated with growth disorders and the tubulin cytoskeleton [17]. The role of the cytoskeleton in the formation of the cell wall is well known and is associated with the location of microtubules (tubulin cytoskeleton) and is a model for studying abiotic stresses [19]. Although the conditions of near space, with the exception of weightlessness, are quite similar to the conditions of life on the surface of the Earth, they are responsible for a wide range of limitations and cause significant changes in the cells of eukaryotic organisms associated with a response to abiotic stress effects [20]. Such changes are highly correlated with cell modifications characteristic of the near space conditions of reduced magnetic field [21,22], increased radiation [23], spatial orientation, microgravity [24] and are associated with cytoskeleton disturbance [25], and gene expression [16,26].

It is well known that the structural features and location of the surface envelopes of cereal kernels, in particular wheat, are a variety-specific character [27,28], sensitive to growing conditions, adverse environmental factors such as high humidity [29]. It was established that the development and morphology of transverse cells of wheat pericarp are highly specific [30], differ depending on the various topographic sites of the kernels that requires a clear definition of the zone location when comparing. The formation of a complex system of kernel tissues, which occurs under altered physical conditions of existence, requires fine regulation and interaction between existing cells and tissues. This can cause local disturbances in osmotic pressure, solute movements at different levels of plant tissue organization, and other disturbances associated with the absence of gravity and only partially compensated by tropic responses [31] on the other influences: light, moisture and the availability of nutrients [32].

The goal of this study was a comparative elucidation of the morphology of surface tissues of wheat kernels formed in space flight conditions, taking into account the peculiarities of cultivation in the “Lada” space greenhouse.

## 2. Materials and Methods

### 2.1. Plant Material

The material for this study was the grains of “Super-Dwarf” wheat (*Triticum aestivum* L.). The “Super-Dwarf” was originally selected in CIMMYT in 1984, by Dr. Bruce Bugbee. It is a hexaploid spring wheat (42 chromosomes), but there is certain amount of germpasm from *T. sphaerococcum* Perc. in “Super-Dwarf” genome. The grains were provided by the Institute of Biomedical Problems of the Russian Academy of Sciences (Moscow) as a material for electron scanning microscopic studies. One control was the seeds harvested from mature wheat plants grown in vessels with soil (initial, parental seeds). Wheat was grown from these seeds in the “Lada” system both in terrestrial conditions and in space station. “Space generation” seeds were obtained from wheat plants, which were cultivated from seed to complete maturation and harvesting in two modules of the “Lada” space greenhouse aboard the ISS RS from the “Rasteniya-2” experiment in 2011. Another control was the wheat seeds obtained in the “Lada” space greenhouse in ground, laboratory conditions (Earth generation). The groups of samples “space”, “Earth” and “parental” consisted of 10 grains per group.

### 2.2. Experiment Conditions

An experiment on growing Super-dwarf wheat was conducted on the territory of the Russian segment aboard the International Space Station (ISS-28/29) by cosmonaut board-engineer S.A. Volkov. The flight experiment included a full cycle of growing wheat from seed to seed and was carried out from 8 August to 17 November 2011. Two blocks of on-board equipment for growing plants “Lada” were used. On 22 November 2011, on a transport manned spacecraft Soyuz TMA-02M (launch on 8 June 2011, landing on 22 November 2011), laying with two plastic bags with dried wheat plants was delivered to Earth.

Plants were grown during the full vegetation cycle in two leafy chambers of the greenhouse, lighting was carried out by fluorescent lamps. Photoperiod for wheat ‒ round-the-clock. Turface© was used as a substrate with the addition of 15 g of Osmokote© prolonged fertilizer per vessel. The moisture content of the substrate was set at the level of 90–96%, the air temperature in the growing zone was about 22 °C.

The temperature and lighting conditions for growing wheat plants in the “Lada” greenhouse in ground laboratory were the same as on board the ISS.

### 2.3. Flight Characteristics

The International Space Station orbits the Earth within the middle of the thermosphere, between 330 and 435 km. The physical parameters of the cultivation of plants in near-Earth orbit corresponded to free fall conditions corresponding to the regime of weightlessness, while the fluctuations in the decrease in the height of the orbit per day ranged from 150 to 200 m. The ISS orbit during the experiment was approximately 353–390 km with temporary short-term periods the appearance of gravity necessary to correct the orbit. We believe that during these periods the results of the entire experiment could not be influenced, since the duration of these periods is not more than a few minutes (On 10 February 2011, the ISS flight altitude was about 353 km above sea level. On 15 June 2011 it increased by 10.2 km and amounted to 374.7 km. On 29 June 2011, the orbit altitude was 384.7 km).

### 2.4. Vegetation Module

During the experiment, the plants were in the “Lada” greenhouse in the atmosphere of a space station consisting of a light module (LM) and a root module (RM) [10] with appropriate modules for water supply and control of operation parameters (Appendix A). Light Module (The LM houses two crew replaceable U-tube fluorescent lamps (Sylvania CF13DS/E/841, Wilmington, MA, USA) and provides the measurement interface for the canopy environment measurement sensor tree. The leaf chamber has a large Lexan window on the front side, covered with a reflective film like the all chamber walls, which can be opened to allow observation and collection of plant grouse. It provides PAR and canopy temperature at three levels (5, 11.5 and 18 cm above the RM) within the canopy). Two fans are mounted on the LM to pull air through the air channel assembly on the top of the RM. This flow pattern provides uniform ventilation through the plant. The gas composition met the requirements for the atmosphere of the cabin and was generally similar to the atmosphere on the surface of the Earth. Cabin temperature range: T = (18.3 to 26.7 °C). Nutrients came to plants from the plug-in module, which is a Root Module (RM). The root container is made of black Delrin (volume—2.5 L). RM includes the nutrients to support multiple generations of plant growth. In the checkout experiment, the RM was filled with a porous (Turface© 1–2 mm) substrate with 15 g of 14-14-14 Osmocote© time-release fertilizer. The bottom of the root container has 20 threaded holes on a 2.54 × 2.54 cm grid to allow variable sensor placement. At “night time”, for the localization of light, the greenhouse chambers were closed by shutters, which could probably cause a local change in gas exchange and temperature in the chamber. In general, the conditions of cultivation in the chambers in near-Earth orbit were similar. The duration of the experiment on obtaining wheat grains from seed to seed and plant cultivation was 90 d and corresponded to the duration of the ground experiment (Appendix A).

### 2.5. Electron Microscopy Technic Protocol

For scanning electron microscopy (SEM), three seeds from parent (soil generation), space generation, and Earths-lab generation samples were collected from normal size seeds (without any deformation and damage) and fixed in a 2.5% glutaraldehyde in 0.1 M Sorenson buffer, pH 7.2. After wash on buffer, samples dehydrated through ethanol series (30% 30’, 50% 30’, 70% 30’, 96% 30’, 2 × 100% 30’). Then CO_2_ for critical-point-dried (Hitachi HCP-2 critical point dryer) was applied. Dry seeds were mounted on a SEM stub with carbon conductive tabs and coated with gold and palladium using an Eiko IB-3 ion-coater (Eiko, Tokyo, Japan). Samples observed under a JSM-6380LA SEM (JEOL, Tokyo, Japan) and a Camscan-S2 SEM (Cambridge Instruments, UK) in the Laboratory of Electron Microscopy (Biological Faculty of Lomonosov Moscow State University).

### 2.6. Statistical Analysis

To estimate the linear and angular parameters, we used photographs from different kernels of each sample with the same magnification, making 10 random measurements (Figure 1). Further processing was carried out in the program Microsoft Excel. Measurements of linear lengths and angular values were measured on photographs of the surface of 3 seeds of different groups. A completely randomized design was used in all experiments. ANOVA and mean separation were carried out using Duncan’s multiple range test. Significance was determined at the 5% level. Significant differences between samples were denoted by letters, where the presence of several letters in the data column means that there are no reliable differences between the corresponding samples, which, for convenience, are duplicated by correction lines. 

## 3. Results

After 90 d of cultivation, the seeds were harvested and analyzed (Figure 1). Grains obtained using a specially engineered space greenhouse, which was equipped with a root module (significantly different from the growth conditions in the ground) in a ground experiment, in orbit and obtained in soil culture had no visual differences. However, it follows from Table 1 that the formed grains in both chambers of the space greenhouse had statistically significant differences in grain weight. We supposed that the physical circumstances of cultivation under conditions of microgravity, weakened geomagnetic field, weightlessness, and specific changes in trophic reactions of plants could cause changes in structural organization and size of cells of the kernel bran and length of the brush hairs. These changes are typical for the action of abiotic factors associated with the availability of moisture, nutrition, and cytoskeletal transformation. For this reason, the task was set to study the bran of the kernels, i.e., the tube and cross cells of the pericarp and epidermal hairs that form the brush.

The fully ripened grains had a typical spherical shape (Figure 2a–c). It is noticeable that the grains obtained under space flight conditions are indeed larger (Figure 2b) than those obtained in the ground experiment (Figure 2a) with “Lada” and in soil culture (Figure 2c).

The histograms (Figure 2d–f) show a significant decrease in the length of the hairs (Figure 2d) and the angle of inclination of the transverse brush hair lines (Figure 2f), while the change in the angle formed by the tip of the brush hair was insignificant (Figure 2e).

These differences are clearly visible (Figure 3a–f), where fragments of brush hairs are shown at a larger magnification, which was used to make measurements (Figure 2). There is a clear similarity in the features of the shape and angle of the inclination of the transverse brush hair lines for grains obtained both in the ground experiment with the “Lada” system (Figure 3a,d) and in parent grains (Figure 3c,f). The angle of inclination of the transverse brush hair lines obtained in orbit is clearly different (Figure 3b,e).

Under the survey of side surface (cheeks) in the middle part of the grains (Figure 1a), grown in the “Lada” greenhouse in the ground experiment (Figure 4a), in orbit (Figure 4b), and the parent grains (Figure 4c), the visual difference was also noticeable. It was manifested both in the arrangement of cells (inserted window) and in the structure of the surface of cell walls, the formation of large and small creases (Figure 4a–c). After calculating the linear dimensions (Figure 1), histograms were obtained (Figure 4d–g).

A significant decrease in the distance between cross cells in the grains obtained on the ISS (Figure 4d) can be seen in histograms 4 d and 4 g with an accompanying decrease in the width of the tube cells (Figure 4g). A decrease in the width of the cross cells was also noted on the surface of the grains obtained in a ground experiment in the “Lada” system, while no differences were observed between the grains from the ISS and the parental ones (Figure 4e). The differences among the specimens in the distance between tube cells of bran were insignificant, although the maximum values were visible in the grains obtained in spaceflight conditions.

## 4. Discussion

A change in the size or shape of an individual organ and surface structure is an indirect reflection of the transformation of cell structures associated with the formation of the cell wall and cytoskeleton [33]. In all cases of deviations from optimal conditions, the plant organism can respond to impact at the molecular, subcellular, cellular, tissue and whole organism levels [34]. Depending on the intensity of the influence, the effect can be both negligible and lead to significant developmental disturbances, and even death. Features of the influence of space flight conditions in near Earth orbit include a number of factors that act both directly and indirectly on various levels of organization of living organisms. Such factors are the presence of the effect of weightlessness, microgravity due to the long fall of the station with short-term correction periods for return of the ISS to orbit (when the effects of gravity are observed), the presence of constant rotation conditions, altered magnetic field conditions, and the associated increased radiation levels that are not typical for ground conditions, and others [32]. A number of these factors can be significantly corrected by the plant organism due to the presence of various tropisms, which made it possible to adjust growth during the development of plant in a changing environment [31]. The mechanisms of these tropisms (phototropism, hydrotropism, and chemotropism) were taken into account when engineering plant growing modules and when choosing crops for such cultivation.

The transformations of the structural organization of “space” seeds (wheat kernels) established in this study did not significantly affect the habit of the plant. However, the revealed violations indicate the presence of an adverse effect. A change in the length of hairs causes a change in the angle of inclination of the thickenings of the cell wall located on their surface in a spiral manner, as well as a slight increase in the angle of tips at the pointed ends of hairs (Figure 2 and Figure 3). It is known that hairs are single cells that make them a convenient target for research. Obviously, their length and surface structure of the formed cell wall can be limited by cell expansion during growth. This process may be contributed by insufficient turgor, associated with a restriction in the uptake of water or high osmotic pressure caused by the accumulation of osmolytes [35]. In addition, growth restriction can be caused by disturbances in the rearrangement of the tubulin cytoskeleton associated with the transformation of cell walls during growth. Moreover, this violation is characteristic of many abiotic stress factors and is manifested in retardation in plant cell growth in various experiments [36,37]. Although it is not possible to identify the source of damage that caused a significant change in size (since there are many factors accompanying space flight conditions in near space), it can be assumed that the observed effect is a very characteristic “earth” effect—a response to the action of one or more abiotic factors [38]. Since the differences between the original parental seeds and the seeds harvested after growing in the “Lada” space greenhouse in ground conditions do not affect the process of seed formation itself, causing a decrease in mass (Table 1), a decrease in the distance between bran cross cells (Figure 4d) and increase in the width of bran cross cells (Figure 4e), it can be assumed that the influence of inappropriate conditions on the vital activity of the aboveground and underground parts of plants in the “Lada” system during the ground experiment has a damaging effect, probably preventing moisture absorption, which is required to supply the cell expansion. When seeds were obtained under similar conditions, but on board the ISS, and although the weight of the “space” seeds was comparable to the weight of the “parental” seeds (Table 1), the differences were revealed in a number of parameters that were not observed for the “Earth” seeds obtained in the “Lada” greenhouse. It can be assumed that it is the specific cultivation conditions during space flight, both related to water uptake and particular features of cell growth and development, they were the cause of the discovered effect. However, these deviations can be circumvented by technically modifying the cultivation system, i.e., they can create an artificial magnetic field, gravity and/or provide protection against radiation. It will be important to eliminate or compensate for each of the probable effects.

Since it is currently not possible to determine what exactly affects the presumed cytoskeleton injuries and the transformation of cell walls, we can only assume that reduced gravity may be the most obvious candidate. Thus, damage to the cytoskeleton, the changes in the expression of genes associated with it, have been established for animal cells, humans, and plants in microgravity conditions [14,22,39]. Another likely damaging agent may be a hypomagnetic (weakened geomagnetic) field, which causes noticeable cytoskeletal disturbances, and also multiple effects associated with changes in chromatin conformation and other epigenetic processes [40,41], which are mediated by correction of gene expression, and other changes characteristic of stressful effects [42,43].

The increased radiation observed in orbit is also a source of multiple damages of cells, disturbances of their growth and development, but we did not take into account the influence of this indicator in our work. However, in some cases it can have a significant effect on the offspring, both in the morphological and genetic aspects, which can extend to appreciable mutations and perceptible changes in the morphology of separate plant organs [23].

It is known that the kernel surface of cereals is sensitive to external influences and is an important indicator depending on the species and variety. The cover layers of the fruit (pericarp) and seed are arranged in such a way that the cells of each layer differ at all stages of development and later play an important protective role [44]. Due to the perpendicular arrangement of relatively elongated cells, they can protect the kernel from mechanical damage and prevent the penetration of moisture (thanks to the hyaline layer located below the seed coat) and pathogens ensuring the defence of both the germ and storage reserves (starchy endosperm). On the other hand, it has been shown that growing conditions can have a significant effect on the relative position of layers and their cell sizes [45]. The results obtained in this work also revealed the sensitivity of these indicators for this wheat variety. Some changes were significant (Figure 4). In order to avoid artifacts associated with uneven development of the layers, we fixed the seeds sideways and precisely examined the central parts of the lateral cheek of the grains. Thus, we revealed differences in such a feature as fine wrinkling of the surface (Figure 4a–c). The difference was particularly noticeable in the location of the tube cells, which were formed less orderly in orbit than during ground growing. A simultaneous decrease in the width of these cells probably indicates that the plant cells experienced either oxidative stress or osmotic stress (associated with insufficient intake of water in this layer) [46]. Other identified changes also allow us to only assume the presence of "internal stress" and build some speculative assumptions. However, like the changes in the hairs described above, the proposed alterations are not essential either for the transfer of genetic information, nor for the formation of a physiologically full-fledged fruit—the kernel. They only suggest certain instability of the conditions of the near space and confirm that, as for the previously studied dicotyledonous plants [47], these conditions do not interfere with the complete development cycle from seed to seed.

Certainly, the revealed differences indicate that space flight conditions are not able to significantly interfere with the cultivation of plants and the production of their full-fledged offspring. The data obtained in these experiments may be relevant for specific cultivation conditions during autonomous human survival in adverse planetary conditions (under water, in caves, in high mountains, in Arctic and Antarctic). However, the revealed differences can be much larger if the experiments are carried out in a near-zero magnetic field, increased radiation, deep space conditions. The differences will indicate the high sensitivity of living systems that despite a long history of existence under various conditions remain sensitive to these common cosmic effects because all living organisms on the planet are largely protected by the Earth’s own magnetic field and its atmosphere.

## 5. Conclusions

Our results indicate that the cultivation and harvesting of wheat in a near space aboard orbital station can cause various disturbances of the structural organization of certain cells of the kernel surface, such as brush hairs, tube and transverse cells. These differences are not an obstacle to proper development of wheat plants and reflect only the consequences of the influence of factors of an abiotic nature, characteristic of any non-optimal cultivation conditions. Understanding the contribution of each stressful as well as stimulating factor will allow us to take into consideration and correct the technology of plant cultivation both on the ground and in the conditions of extraterrestrial greenhouses.

## Figures and Tables

**Figure 1 life-09-00081-f001:**
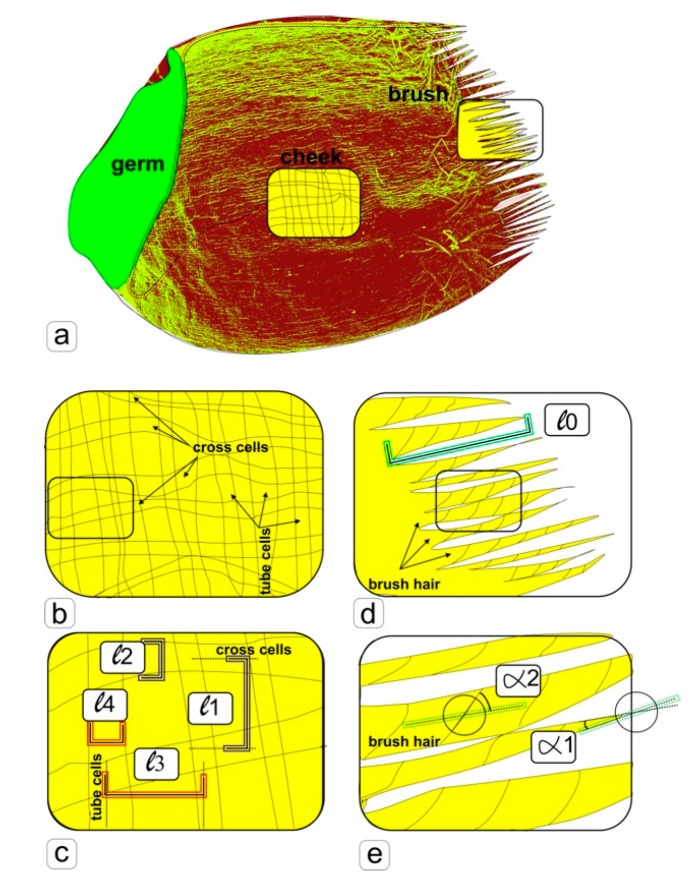
Scheme of morpho-anatomical analysis of images of surface tissues of a wheat kernel obtained by scanning electron microscopy. **a**—the structure of a wheat kernel (one-seeded fruit), and zones, in which measurements were taken (cheek, brush), **b**—the allocation scheme of the envelopes of mature wheat pericarp (cross cells and tube cells), **c**—the scheme of measuring for analyzing the distances between cells (l1 and l3) and the width of cells (l2 and l4), **d**—the arrangement scheme of structural elements of wheat brush hairs, **e**—scheme for measuring of angular parameters for analysis.

**Figure 2 life-09-00081-f002:**
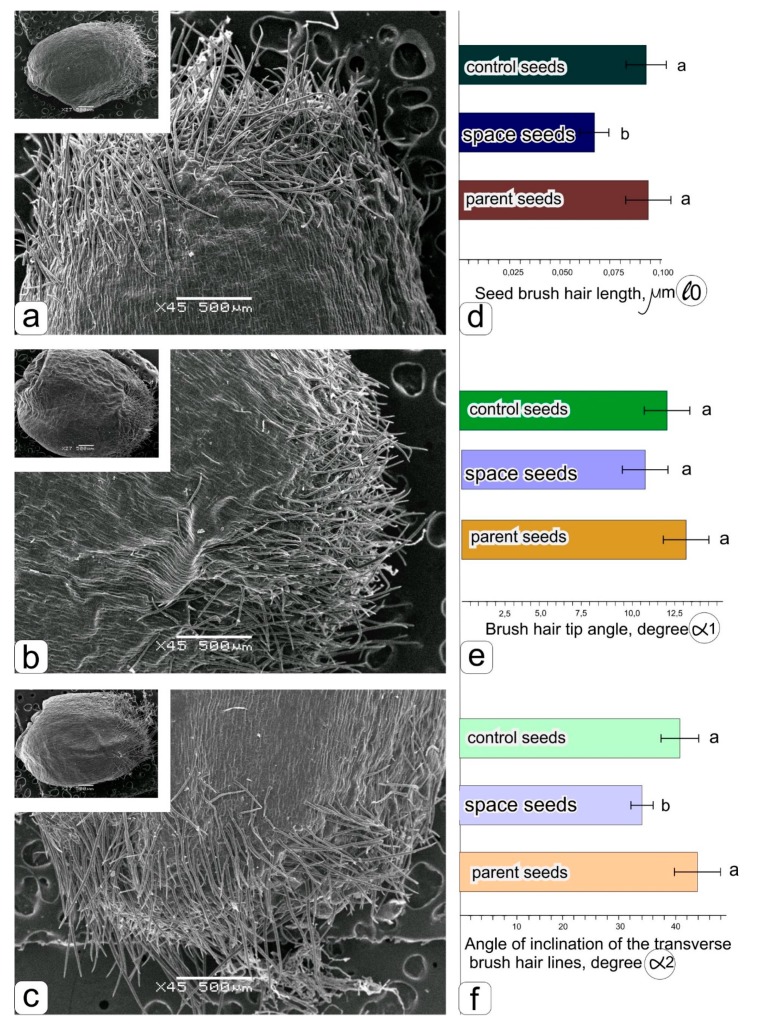
The surface of the kernels and brush hairs of the grains obtained: when cultivated in the space greenhouse “Lada” in ground conditions–**a**, in the “Lada” system in an orbit–**b**, parental grains harvested from in soil grown wheat–**c**. Histograms obtained as a result of statistical processing of data on measuring the length of brush hairs–**d**, the angle of the tip of the hair–**e**, the angle of inclination of transverse brush hair lines–**f**.

**Figure 3 life-09-00081-f003:**
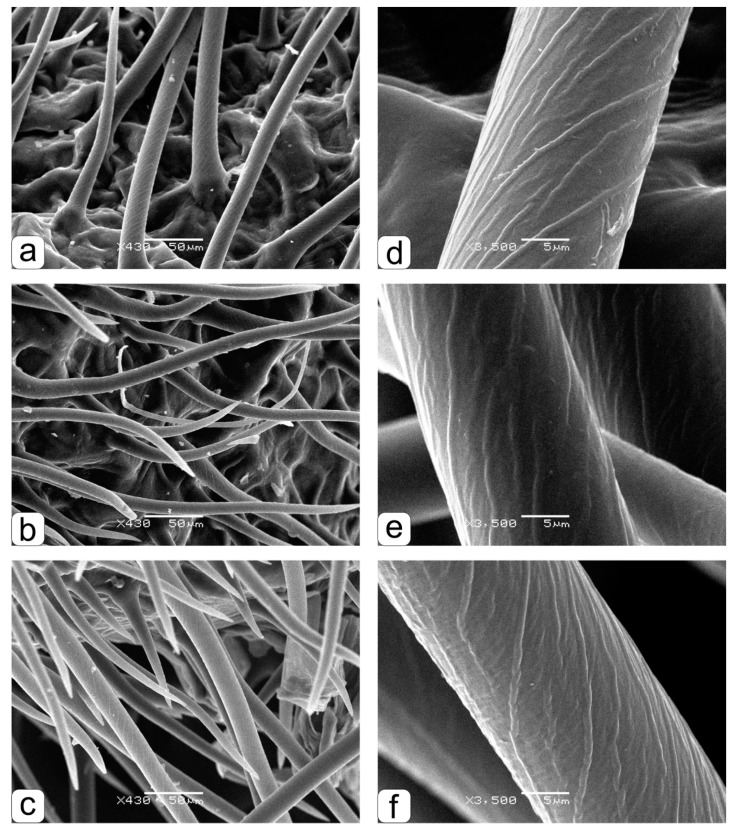
The structural organization of brush hairs of the kernels obtained during cultivation in the space greenhouse “Lada” in ground conditions–**a**, in the “Lada” system in an orbit–**b**, parental grains harvested from in soil grown wheat–**c**. Ultrastructure of hairs with helical transverse lines in kernels obtained in ground conditions–**d**, in orbital conditions–**e**, in parental grains obtained in ground soil culture–**f**.

**Figure 4 life-09-00081-f004:**
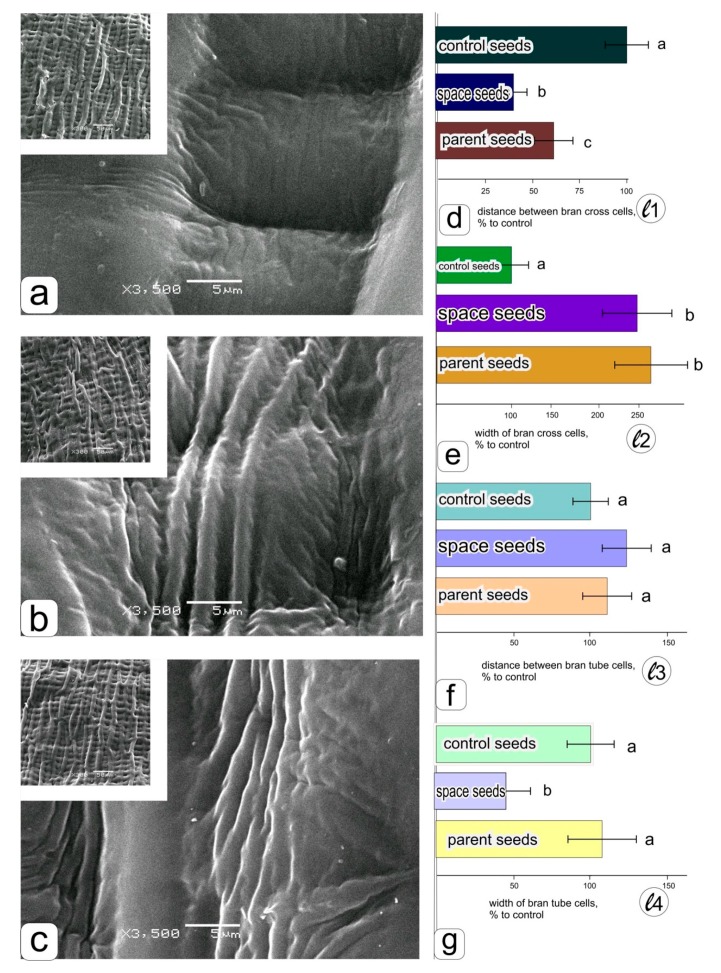
Ultrastructural organization of the kernel coats of the cheeks of wheat grains obtained by growing in the “Lada” system in ground conditions–**a**, in the “Lada” system in orbital conditions–**b**, the surface of parental grains obtained in ground soil culture–**c**. Histograms obtained as a result of statistical processing of data on measuring the distance between bran cross cells–**d**, the width of bran cross cells–**e**, the distance between bran tube cells–**f**, the width of bran tube cells–**g**.

**Table 1 life-09-00081-t001:** Characteristics of Super-dwarf wheat grains obtained after 90 d of cultivation in the “Lada” space greenhouse under the conditions of a ground-based laboratory experiment, an orbital flight (ISS), in near space (dry weight), and the original parent seeds grown in soil culture.

Variants	Weight 1000 Grains, g
**Module of space greenhouse Lada №23**	35.28 ± 0.5a
**Module of space greenhouse Lada №24**	34.26 ± 0.5b
**Module of space greenhouse Lada ground control**	31.0 ± 0.5c
**Parental ground control grown in soil**	33.4 ± 0.6b

Variants labeled with same letters do not differ significantly by the Duncan’s test (α = 0.05).

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
