# Peer review of "Wheat Space Odyssey: “From Seed to Seed”. Kernel Morphology"

_life, 2019, doi:10.3390/life9040081_

Round 1
Reviewer 1 Report
The manuscript described the observed differences in structural features on kernel surface of super-dwarf wheat grown under two conditions: near Earth orbit condition and ground condition. The authors suggested the observed differences were due to disturbance in cytoskeleton affected by yet to be identified factors. However, authors failed to provide any quantitative measure of cytoskeleton to support the hypothesis. In addition, the authors did not establish any links from the observed kernel surface structure to further physiological processes of the wheat offspring, undermining the significance of this work.
The section “2.1 plant material” was written in a very confusing way, and did not help explaining what “control seeds” and “MKS seeds” mean later in the figure 2 and 4.
The description on statistical analysis was not described clearly. It should include which statistical test was used for calculating P value, and what the error bars represent. In addition, the letters indicating different significance group in figure 2 and 4 are very redundant. For example, in figure 2e, all three groups contain letters a, b and c, though presented in varying order, indicating they were assigned within the same significance group. why not only use one letter to indicate not significant difference among them?
What does “round-the-clock lighting regime” mean exactly?
What does “However, it can be assumed that this can be easily overcome if necessary.” refer to?
Other minor concerns:
Please provide the complete information for the authors’ affiliation.
In the second sentence of Introduction, “theraploid wheat” should be “tetraploid wheat”
Reviewer 2 Report
The manuscript has not been formatted for the life style, so please, use the temple.
The authors report a study on the kernel obtained from wheat growth on the ISS and on ground greenhouse. Morphology studies were carried out using Electron Microscopy.
In the statistical section of the materials and methods report the statistical analysis description and One-way ANOVA should be used for the determination of statistical differences.
Regarding the cultivation are there any information on the biomass production and number of seeds per plant?
Do grains reach the fully maturity stage?
Provide a photo of the plants during growth, if available.
Table 1. Please use dots for decimals instead of comma. Report the letters of significance among means.
Provide the section of conclusions at the end of the manuscript reporting the main message of this work and knowledge acquired.
Reviewer 3 Report
Please find my notes in text. Also I believe that manuscript must be completed by brief conclusions.

Reviewer 4 Report
The paper is interesting and suitable for the journal, but:
a) Authors have not used Microsoft Word template.
b) Authors should include image/drawing of vegetation module (2.4. Section). On earth and in space. With text only, It's hard.
c) Statistical Analysis (2.6 Section). Is it ANOVA? What is the multiple comparison test? All this should be better described. Also in figures.
Round 2
Reviewer 1 Report
The revised version of the manuscript has been marginally improved over the original version. The authors failed to address several issues I had pointed out in my previous comments.
First, there was still no clear explanation what “MKS seeds” was referring to.
Second, the letters used for denoting significant differences should be simplified in Fig. 2 and Fig. 4. For example, in Fig. 2d, hair length for MKS seeds is significantly smaller than the control seeds and the parent seeds and there is no difference between the latter two, therefore letter a can be assigned to the control seeds and the parent seeds and letter b to MKS seeds. The letter c is not needed here and should be removed, as there is no intermediate group that is not significantly different from either a or b. In the case of Fig. 2e, only one letter is needed, rather than use a, b and c in shuffling orders, as there is no difference between the groups.
Third, the Introduction must be revised to provide a clear rationale outlining the background and challenges leading to the objectives of the current study. The observation that morphological changes occurred in the space grown wheat kernels was interesting, but the interpretation should not be stretched too far, especially not early in the Introduction. The authors provided some reasonable connections between kernel morphology, cell wall, and stress responses in the letter. I suggest the authors should add these in the revised manuscript.
In addition, the goal of this study was not written clearly either. The authors listed three means to achieve a goal, but did not elaborate on what the goal of this study really was. I suggest the authors need to implement the rationale of the study and write down a clear research goal in the revised Introduction.
Line 262 to 263, “That is why a round-the-clock lighting regime is used…” I could not comprehend the reason the authors were referring to for using a round-the-clock lighting regime.
Line 280 to 282, “Since the differences between the original parental seeds and the seeds harvested after growing in the “Lada” space greenhouse in ground conditions are insignificant for all the parameters studied…” Aren’t that distance between bran cross cells (Fig. 4d) and width of bran cross cells (Fig. 4e) have significant differences between control seeds and parental seeds?
Avoid using phrases like “obviously”, “it is obvious that”, and “it is well known that”, because they undermine objectivity of the manuscript and the subjects discussed may not be obvious to readers.
Author Response
Thank you for your thorough review.
In order to eliminate the comments, changes were made in Fig. 2 and Fig. 4, and “MKS seeds” replaced by “space seeds” as described.
Also, corrected illustrations are inserted into the text in which corrected symbols in the histograms are entered.
The introduction was amended to remove fragments of the redundant broad interpretation and reformulated the goal.
The phrase about “round-the-clock lighting” was deleted as redundant technical information.
A significant comment on the interpretation of the results has been changed, references to the figures have been introduced into the text, and a fragment of the discussion has been replaced. Indeed, the noted differences did not allow us to make the initially proposed assumption.
The text has been corrected to eliminate the unjustified use of introductory words.
Please see the attachment.

Round 3
Reviewer 1 Report
The latest revision has substantially improved the manuscript, and has sufficiently addressed my previous comments.